# Identification of Antimicrobial Peptides from the Microalgae *Tetraselmis suecica* (Kylin) Butcher and Bactericidal Activity Improvement

**DOI:** 10.3390/md17080453

**Published:** 2019-08-01

**Authors:** Fanny Guzmán, Genezareth Wong, Tanya Román, Constanza Cárdenas, Claudio Alvárez, Paulina Schmitt, Fernando Albericio, Verónica Rojas

**Affiliations:** 1Nucleo Biotecnología Curauma, Pontificia Universidad Católica de Valparaíso, Valparaíso 2373223, Chile; 2Instituto de Biología, Pontificia Universidad Católica de Valparaíso, Valparaíso 2373223, Chile; 3Laboratorio de Fisiología y Genética Marina (FIGEMA), Centro de Estudios Avanzados en Zonas Áridas (CEAZA), Coquimbo 1781421, Chile; 4Facultad de Ciencias del Mar, Universidad Católica del Norte, Coquimbo 1781421, Chile; 5Department of Organic Chemistry and CIBER-BBN, Networking Centre on Bioengineering, Biomaterials and Nanomedicine, University of Barcelona, 08028 Barcelona, Spain; 6School of Chemistry, University of KwaZulu-Natal, Durban 4001, South Africa

**Keywords:** antimicrobial peptides, microalga, *Tetraselmis suecica*, alanine scan, lysine analogs, alpha helical secondary structure

## Abstract

The outburst of microbial resistance to antibiotics creates the need for new sources of active compounds for the treatment of pathogenic microorganisms. Marine microalgae are of particular interest in this context because they have developed tolerance and defense strategies to resist the exposure to pathogenic bacteria, viruses, and fungi in the aquatic environment. Although antimicrobial activities have been reported for some microalgae, natural algal bioactive peptides have not been described yet. In this work, acid extracts from the microalga *Tetraselmis suecica* with antibacterial activity were analyzed, and de novo sequences of peptides were determined. Synthetic peptides and their alanine and lysine analogs allowed identifying key residues and increasing their antibacterial activity. Additionally, it was determined that the localization of positive charges within the peptide sequence influences the secondary structure with tendency to form an alpha helical structure.

## 1. Introduction

Microalgae are ubiquitous unicellular eukaryotic photosynthetic organisms distributed in quite diverse and extreme environments. They are especially valuable due to their high content of compounds with different biological activities, including both complex organic compounds and primary and secondary metabolites, such as phytopigments (xanthophylls and carotenoids), polyunsaturated fatty acids (PUFAs), phenolic substances, docosahexaenoic acid (DHA), vitamins, carbohydrates, tannins, terpenoids, and peptides, among others [1,2,3,4].

Marine microalgae have been target organisms in the search for new antibiotic molecules required to confront the resistance to antibiotics that impacts the efficacy of conventional therapies against bacterial infections in human and animals [5,6,7,8]. These microorganisms have developed tolerance and defense strategies to resist the exposure to bacteria, viruses, and fungi pathogens [1]. For this reason, microalgae extracts have been tested on cultures of these organisms. The antimicrobial activity of these extracts has been associated to different molecules such as fatty acids, phenolic compounds, tannins, terpenoids, polysaccharides, indole, acetogenin, halogenated hydrocarbons, amides, alkaloids, and inhibitory enzymes [1,9,10,11]. 

Peptides derived from microalgae have been mainly obtained by enzymatic digestion using proteolytic enzymes. They have been mostly obtained from protein hydrolyzates of *Chlorella vulgaris*, *Chlorella ellipsoidea*, *Tetradesmus obliquus*, *Navicula incerta*, and *Nannochloropsis oculata* [12,13,14,15,16]. The resulting bioactive peptides have shown to possess different biological functionalities, such as antioxidant, anticancer, antihypertensive, and antimicrobial activities, with beneficial health effects and potential therapeutic applications [17]. 

*Tetraselmis suecica* is a marine green microalga widely used in aquaculture as live food for bivalve mollusks, crustaceans, and rotifers; moreover, it has been proposed as a source of vitamin E for mammals and humans. Some antibacterial activity has been also described against *Staphylococcus*, which has been attributed to terpenes [18]. In addition, this microalga has been shown to inhibit the growth of bacteria such as *Vibrio anguillarum*, *Aeromonas hydrophila*, *Aeromonas salmonicida*, and *Lactobacillus* sp., but this effect has not been associated with any specific compound [19,20].

In this report, the antibacterial activity of *T. suecica* was determined, and for the first time the antimicrobial peptides (AMPs) involved were identified. Herein, we report the characterization and chemical synthesis of three *T. suecica* AMPs active against Gram-positive and Gram-negative bacteria. The peptide with the highest activity was subjected to alanine scanning that allowed identifying the contribution of each amino acid residue to the peptide’s functionality, stability, and structure [21,22]. After that, in order to improve the antibacterial activity and peptide solubility, some residues were replaced by basic amino acid residues, such as lysine, which have been reported to increase the antimicrobial action [23,24]. Alanine and lysine analogs of the selected peptide resulted in enhanced antibacterial activity without cytotoxic effects to eukaryotic cells.

## 2. Results

### 2.1. Characterization and Antimicrobial Activity of Acid Extracts from Tetraselmis suecica

Microalga was submitted to acid extract, and purified by C-18 reverse-phase column chromatography to obtain fractions with differential hydrophobicity, using an elution gradient of 5%, 40%, 80%, and 100% acetonitrile (ACN) in 0.01% trifluoroacetic acid (TFA) in water. This procedure modified from Mitta [25] has been used by our group in the search of peptides from other natural products, with good results [26,27].

Among all elutions, the 40% ACN eluted fraction showed both the highest protein concentration and antibacterial activity. Furthermore, peptides below 10 kDa were observed in the 40% ACN eluted fraction by Tris-Tricine SDS-PAGE (Figure 1).

The antibacterial activity of the 40% ACN eluted fraction was tested against three Gram– and four Gram+ bacterial strains. Results show that the 40% ACN eluted fraction was active against *Escherichia coli*, killing 96% of bacteria after incubation with 0.5 µg/µL of the peptide extract. Antibacterial activity against methicillin-resistant *Staphylococcus aureus* (MRSA) and *Bacillus cereus* Gram+ strains was also observed. The strongest bactericidal activity recorded was against *B. cereus*, where the peptide extract killed 100% of the bacteria after being incubated with 0.5 µg/µL of the peptides extract (Table 1). 

### 2.2. Peptide De Novo Sequence and Antibacterial Activity

The 40% ACN eluted fraction was further purified until homogeneity by reverse phase high-performance liquid chromatography (RP-HPLC) in a continuous 0%–80% ACN gradient in 0.01% TFA in water for 20 min at 1 mL/min. HPLC fractions were collected, lyophilized, and subsequently characterized by Matrix-Assisted Laser Desorption Ionization-Time of Flight (MALDI-TOF) mass spectrometry and by MS/MS mass spectrometry to determine de novo sequences of the peptides.

Results showed 24 sequences between 3 to 8 amino acid residues in length. A Blast search in NCBI [29] within *Tetraselmis* genera (taxonomy id: 3164) showed that 9 of the 24 sequences were contained within the available sequences of *T. suecica* with identities from 40% to 100%; other sequences have similar identities within these genera, and 4 of them did not have any matching (Appendix A). 

The identified peptides were then chemically synthesized by Fmoc solid-phase strategy with amidated C-terminal, and their antimicrobial properties were evaluated. Synthetic peptides were tested against *E. coli* and *Micrococcus luteus* at 50 μM peptide concentration (Figure 2). The most active peptides were AQ-1756, AQ-1757, and AQ-1766; remarkably, AQ-1766 exhibited an antibacterial activity similar to the control peptide BTM P1 (Figure 2). The peptides showing the highest antibacterial activity were chosen for further analysis.

### 2.3. Structural Characterization and Cytotoxicity Analysis of Antimicrobial Peptides of T. suecica

Analysis of the secondary structure of the peptides by circular dichroism (CD) showed that AQ-1756 had a pattern corresponding to random coil, while AQ-1757 had one of a beta turn, and AQ-1766 had one of a helical structure with some distortion associated to the presence of a tryptophan residue (Appendix A). 

After one hour of treatment, none of the peptides exhibited cytotoxic activity on human embryonic kidney cells (HEK293) at concentrations ranging from 12.5 to 100 µM by MTS cell viability assay [30]. However, cell viability decreased to 75% after 24 h of treatment. None of the three peptides showed a significant difference with respect to the untreated cells at the lowest tested concentration. In addition, AQ-1766 showed the lowest toxicity, without significant difference with respect to the negative control at 50 µM (Figure 3, top panel). 

Similar results were obtained by flow cytometry analysis, given that no significant differences were detected between untreated and peptide treated cells. After exposure to AQ-1756, AQ -1757, and AQ-1766, the percentage of viability of HEK293 cells were 88.13%, 88.40%, and 91.98% respectively, compared with 98.66% for untreated cells (Figure 3, bottom panel).

### 2.4. Characterization and Activity of Analog Peptides

Based on antibacterial and cytotoxic activities, the peptide AQ-1766 was selected to perform additional tests. In order to evaluate the role played by each residue in the structure and antimicrobial activity of AQ-1766, analog peptides were synthesized by replacing each amino acid residue by alanine; then, residues susceptible to modification were replaced by lysine and chemically synthesized (Table 2).

#### 2.4.1. Antibacterial Activity of AQ-1766 Alanine Scan Analogs

The antibacterial activity of alanine scan analogs was tested against Gram– (*E. coli*, *Salmonella typhimurium*, and *Pseudomonas aeruginosa*) and Gram+ (*B. cereus*, methicillin-resistant *S. aureus* (MRSA), *Listeria Monocytogenes*, and *M. luteus*) bacterial strains. Bactericidal peptide BTM-P1 derived from *Bacillus thuringiensis* at 30 µM was used as positive control showing 0% of bacterial survival in all assays. 

In general, AQ-1766 was more effective against Gram+ than Gram– bacteria, with minimal bactericidal concentration (MBC) values between 40 and 50 µM (Figure 4 and Table 3). In addition, AQ-1766 showed low activity against *S. typhimurium*, *P. aeruginosa*, and *L. monocytogenes*, and it was not possible to determine MBC_50_ at the concentrations used nor to perform the statistical analysis (Figure 4 and Table 3). Some alanine scan analogs showed a clear decrease in antimicrobial activity, including the substitution of tryptophan by alanine in the second and seventh position (AQ-2998 and AQ-3003, respectively), and the substitution of histidine by alanine in the eighth position (AQ-3004). On the contrary, the substitution of tyrosine by alanine in the fourth position (AQ-3000), the substitution of threonine by alanine in the fifth position (AQ-3001), and the substitution of methionine by alanine in the sixth position (AQ-3002) produced an increase in antimicrobial activity, as can be seen in the survival curves (Figure 4), and from them, the 50% survival calculation expressed as the minimal bactericidal concentration 50 (MBC_50_) (Table 3). Moreover, alanine substitutions had no significative effect on the activity against *P. aeruginosa*, showing a similar behavior to the original peptide (Table 3).

#### 2.4.2. Antibacterial Activity of AQ-1766 Lysine Scan Analogs 

The evaluation of lysine scan analogs showed antibacterial activity improvement against all the bacteria tested with respect to the reference peptide, as can be seen in the survival curves (Figure 5) and the MBC_50_ values (Table 4). The substitution of leucine by lysine in the first position (AQ-3369) and substitution of tyrosine by lysine in the fourth position (AQ-3370) exhibited the highest antibacterial activity. In addition, the substitution of methionine by lysine in the sixth position (AQ-3372) increased the antibacterial activity, but it was not better than the substitution by alanine (AQ-3002). As in the alanine scan, it was not possible to determine the significance of the difference against *S. typhimurium* and *L. monocytogenes* of the original versus lysine analog peptides, but a reduction in the MBC_50_ was observed. In the case of *P. aeruginosa*, although it was not possible to determine MBC_50_ at the used concentrations, the survival curves showed a decrease in bacterial survival percentage.

#### 2.4.3. Secondary Structure Analysis of AQ-1766 Alanine and Lysine Scan Analogs

Secondary structures of AQ-1766 and their alanine and lysine analogs were analyzed by CD at 1 mM peptide concentration on 30% trifluoroethanol (TFE) as solvent. The CD spectra of AQ-1766 and alanine scan peptides showed a tendency to form a β-sheet structure, with a minimum of molar ellipticity between 210–220 nm and a maximum of molar ellipticity between 195–200 nm [32]. Structures of this type have been described in other short-chain peptides, such as 7-residue alanine oligomers [33], where it was suggested that interactions of the peptides with the solvent as well as the intramolecular interactions are key to the conformation of these peptides. In addition, the alanine substitutions of tryptophan in the second position completely modified the CD spectra, showing an undetermined structure (Appendix A). 

The lysine analogs exhibited a greater CD spectra modification than alanine analogs (Figure 6). The secondary structure of the peptides showed a helical tendency, with the two minima and the maximum observed at the wavelengths corresponding to these structures [32], 208 and 222 nm for the minima and 193 nm for the maximum, but the spectra appeared distorted by the presence of aromatic residues [34].

#### 2.4.4. Cytotoxicity Analysis of AQ-1766 Alanine and Lysine Scan Analogs

Cytotoxicity of AQ-1766 and its alanine and lysine analogs were evaluated in HEK293 cells by the MTS assay [30]. Both types of analogs gave results similar than obtained with peptide AQ-1766, without cytotoxic effects after exposure to 50 μM of the peptides for one hour (data not shown).

## 3. Discussion

We report here the existence of AMPs from the microalgae *T. suecica*, with bactericidal effect against Gram− and Gram+ human pathogenic bacteria. The antibacterial properties of protein hydrolysates were described for the microalgae *Dunaliella salina* [35] and *Chlorella sorokiniana* [36], but the specific peptides responsible for this activity were not determined. Thus, to our knowledge, this is the first report of antimicrobial activity associated with peptides in microalgae.

The antibacterial activity of microalgal extracts has been previously reported, but it was not attributed to peptides [37], despite it being reported that microalgal peptides may exhibit different kinds of bioactivities such as antioxidant [38,39,40], antihypertensive [39], anticancer [41], and liver protection effects [12]. Nevertheless, no AMPs from microalgae have been described so far.

Bioactive peptides from microalgae have been commonly obtained by protein hydrolysis of cell extracts. These peptides are inactive and encrypted until liberation by enzymatic hydrolysis [42]. In this work, peptides naturally present in the microalgae *T. suecica* were obtained using a purification procedure that minimized hydrolysis. Furthermore, the eluted fraction after Sep-Pak chromatography displaying antibacterial activity was the 40% ACN, similar to several reports where AMPs were purified [25]. 

De novo sequencing by mass spectrometry allowed us to define the sequences of 24 peptides that were chemically synthesized, whose antimicrobial properties were evaluated. Three of these synthetic peptides showed significant antibacterial activity. Among them, the peptide named AQ-1766 showed the highest effect on Gram− and Gram+ bacterial strains. The peptides had no cytotoxic effect on human cell line HEK293 at the concentrations used in the antibacterial assays. Sequences of two antibacterial peptides, named as AQ-1756 and AQ-1766, showed identities of 80% and 60% with *T. suecica* reported proteins, respectively. Moreover, sequence AQ-1757 showed an 80% identity with a *Tetraselmis chui* protein. The *T. suecica* genome has not been reported yet; therefore, it was not possible to identify the other peptide sequences. Neither have been described peptides liberated by enzymatic hydrolysis nor their primary sequences of proteins from the *Tetraselmis* genera [41]. 

Alanine substitution has been commonly used to probe the contribution of each residue in the structure and antimicrobial activity of peptides because the substitution of a side chain residue by methyl groups provides an effective strategy for detecting the side chains responsible for secondary structure and activity [21,43]. In this study, AQ-1766 and its alanine analogs showed similar secondary structures, with a slight tendency to form helical structures, but with distortions in the CD spectra probably due to the presence of the aromatic residues, in particular tryptophan, as previously described for other peptides [34,44]. Nevertheless, the replacement of residues in positions two, three, seven, and eight (Trp, Phe, Trp, and His, respectively) significantly decreased the antimicrobial activity. The presence of tryptophan is essential in the antibacterial function, since this residue may participate directly in the insertion of the peptide in the bacterial plasma membrane through interaction and stabilization with its surface, as observed in other AMPs [45,46]. Interestingly, the substitution of histidine residue in the carboxy terminal of the peptide greatly diminished its activity. Future work is intended to establish the mechanism of action of *T. suecica* peptides.

On the other hand, substitutions of leucine, tyrosine, threonine, and methionine by alanine at positions one, four, five, and six of the peptides AQ-2997, AQ-3000, AQ-3001, and AQ-3002, respectively, produced an increase of their antimicrobial activity, whereby these positions were chosen to make replacements with lysine. These analogs exhibited interesting properties. Firstly, their antibacterial activity was higher than the one of the reference peptide (AQ-1766), particularly against Gram+ bacteria. This effect can be related to the increase of the peptide’s positive net charge, associated with the lysine side chain, which is able to interact with the negatively charged bacterial membranes [43,47]. Secondly, a notorious change in the CD spectra of the lysine analogs was observed with respect to the AQ-1766 peptide and their alanine counterparts, which also suggests the lysine influence in the secondary structure. Prediction of secondary structure is limited due to the short sequences of the peptides. However, according to the structure prediction by PEPFOLD3 Peptide Structure Prediction server [48], all peptides have a trend to form helical structures (Appendix A). In a helical wheel representation, the peptides with the highest activity, AQ-3369 and AQ-3370, exhibited two basic residues in the same face of the helix, which could contribute to stabilize the structure (Appendix A). Therefore, lysine substitutions improved the antibacterial activity of the reference peptide by both secondary structure stabilization and increase in the positive net charge of the peptides augmenting their interaction with the bacterial membrane. 

The activity of the 40% ACN extract seems to be higher than that of the synthesized peptides (Table 1); however, the concentration of the peptides used ranged from 0.01 µg/µL (10 µM) to 0.06 µg/µL (50 µM) for the different peptides, which is at least one order of magnitude lower than in the extract. In addition to this, having a pure peptide whose structure is known allows the assays to be easily reproducible and scalable, which does not occur with natural extracts.

Future work will allow defining the action mechanism of these AMPs. It is worth noting that the lysine analogs had higher solubility and were obtained at higher yield (data not shown), which are desirable features in terms of production and application. 

Overall, the identification and functional characterization of antibacterial peptides from the microalgae *T. suecica* is a contribution to the development of new alternatives for the control of infectious diseases. One advantage of antibacterial peptides from microalgae is that they could be administered as diet supplements for human consumption or as medicinal feed in animal production, considering that microalgae and its derivatives are currently used for such purposes [3,13,49]. Microalgal AMPs is an area scarcely explored, and since more than 30,000 species of microalgae have been already described, this new source of antimicrobials could contribute to palliate the increase in the appearance of bacterial infections and the serious problem of antibiotic resistance that affects the effectiveness of therapies in human and animal health [50,51].

## 4. Materials and Methods 

### 4.1. Tetraselmis Suecica Culture

The marine alga *T. suecica* (strain CCAP904) was obtained from the National Center for Marine Algae and Microbiota (NCMA) at the Bigelow Laboratory, East Boothbay, Maine, USA. 

*T. suecica* was cultured in F/2 medium [52] at 21 ± 0.5 °C and controlled lighting (40 W), in accordance with Paniagua et al [53]. Cultures were prepared from axenic stocks and were progressively scaled until a final volume of 20 L.

### 4.2. Production of Acid Extracts

Acid extracts were obtained by a method adapted from Mitta et al [25]. Briefly, algae were collected by centrifugation at 4690× *g* for 20 min at 4 °C, and the microalgal pellets were dissolved in cold 5% acetic acid (AcOH) plus 1× of the halt protease inhibitor cocktail, EDTA-free (Thermo Scientific, Waltham, MA, USA). Algae lysis was performed by homogenization with zirconium spherical particles (0.44 mm) in a FastPrep-24^®^ tissue homogenizer (MP Biomedicals, Irvine, CA, USA), (3 cycles of 35 s at a power of 5 m/s at 40,000 °C). Then, the homogenate was centrifuged at 15,000× *g* for 30 min, and the supernatant was stored at −80 °C until use.

### 4.3. Enrichment and Characterization of T. suecica Extracts

Extracts were partially purified by solid phase hydrophobic interaction chromatography in a Sep-Pak C-18 LiChrolut^®^ RP-18 Column (Merck, Darmstad, Germany) with an elution gradient of 5%, 40%, 80%, and 100% acetonitrile (ACN) in 0.01% TFA in water. Eluted fractions were lyophilized and reconstituted in 0.01% acetic acid for antibacterial assays. The post Sep-Pak eluted fractions were characterized regarding protein concentration and antibacterial activity.

The 40% ACN eluted fractions, which presented both the major protein content and antibacterial activity, were partially purified by RP-HPLC (Jasco Analytical Instruments, Tokyo, Japan) with an Atlantis^®^ dC18, 3 µm × 4.6 mm × 150 mm (Waters) column. Chromatography was performed in a continuous gradient 0%–80% ACN in 0.01% TFA in water for 20 min at 1 mL/min. UV absorption was read at 214 nm. HPLC fractions were concentrated by rotoevaporation in a SpeedVac Concentrator Savant SPD1010 (Thermo Scientific, Waltham, MA, USA) and lyophilized.

### 4.4. Characterization of the Enriched Peptide Fractions by Gel Electrophoresis

Total protein concentration in post Sep-Pak 40% ACN eluted fraction was estimated using the bicinchoninic acid (BCA) assay (Pierce Protein Assay) in accordance with manufacturer’s instructions (Thermo Scientific, Waltham, MA, USA). The molecular mass of the sample was estimated by Tricine/SDS/PAGE [54] using a 16.5% separating gel, 10% spacer, gel and 4% stacking gel in the presence of 2-mercaptoethanol, and the protein/peptide bands were stained with Coomassie Brilliant Blue (Sigma-Aldrich, St Louis, MO, USA). 

### 4.5. Antibacterial Assays 

Antibacterial activity was determined against different species of the Gram-negative and Gram-positive human pathogenic bacteria: *Escherichia coli* ML35 (ATCC 25922), *Salmonella typhimurium* (ATCC 14028), *Pseudomonas aeruginosa* (ATCC 27853), *Bacillus cereus*, (ISP B7/13), *Staphylococcus aureus* (ATCC 25933), *Listeria monocytogenes* (ATCC 19115), and *Micrococcus luteus* (ATCC 9341). 

The minimum bactericidal concentration (MBC) was evaluated in 96-well microtiter plates as reported [43,55]. Bacteria were cultured overnight in trypticase soy broth (TSB) and diluted until reaching an optical density (OD_600_) of 0.4 to 0.7. Cultures containing 1 × 10^7^ colony-forming units per milliliter (CFU/mL) were exposed for 1 h at 37 °C to different concentrations of the 40% ACN eluted fractions or to each concentration of the synthetic peptides using 1% TSB in 10 mM HEPES buffer. Following the exposure, bacteria were subjected to seven ten-fold serial dilutions and incubated for 16 h at 37 °C in fresh TSB media. Surviving bacteria were quantified as colony-forming units per milliliter (CFU/mL) for each elution and peptide concentration. 

Bacteria without extracts or peptides were used as negative control, while bacteria incubated with the bactericidal peptide BTM-P1 derived from *Bacillus thuringiensis* [56] at a concentration of 30 µM were used as positive control. All assays were carried out in triplicate. To verify the bactericidal activity of the extracts, the treated bacteria were plated on 1.5% trypticase soy agar (TSA), incubated overnight, and the bacterial growth was evaluated through the formation of colonies.

In the antibacterial assays, the values of bacterial survival in CFU/mL were transformed to log_10_ and normalized with respect to the negative control to obtain the percentage of bacterial survival. These are the values presented in the respective graphs, and the MBC 50 for each peptide was interpolated from the graphics at 50% of bacterial survival.

### 4.6. Mass Spectrometry

The 40% ACN eluted fractions were analyzed by RP-HPLC and subjected to mass spectrometry in a MALDI-TOF Microflex equipment (Bruker Daltonics GmbH, Bremen, Germany). For this, samples were suspended in 0.1% v/v formic acid or 3% v/v methanol and mixed 1:1 v/v with 10 mg/mL of the α-cyano-4-hydroxycinnamic acid (CHCA) matrix in ACN/0.1% formic acid or suspended in 1:2 v/v ACN/0.1% formic acid and mixed with 10 mg/mL of the sinapinic acid (SA) matrix. The acquisition of the spectra was performed in positive ion mode through detection by reflection for samples in CHCA matrix (interval 0–5 kDa) and through lineal detection for samples in SA matrix. The spectrum corresponded to the sum of 10 sweeps of 30 laser impacts for samples in CHCA matrix and to the sum of 40 laser impacts for samples in SA matrix, for different points taken at random.

### 4.7. LC-ESI-MS/MS Mass Spectrometry

To determine the de novo sequence of the peptides, the MS/MS spectra were analyzed by the FlexAnalysis program version 3.0 (Bruker Daltonics GmbH, Bremen, Germany). The detection of the m/z signals was performed with the sophisticated numerical annotation procedure (SNAP) algorithm for samples in CHCA matrix (signal/noise ratio of 6), and with the centroid algorithm for samples in the SA matrix (signal/noise ratio of 3).

### 4.8. Peptide Synthesis and Purification

Peptides were synthetized using 9-fluorenylmethoxycarbonyl (Fmoc) solid-phase strategy in a Rink amide resin (Iris Biotech GmbH, Marktredwitz, Germany) (0.59 meq/g). Fmoc deprotection was performed with 20% v/v piperidine in *N*,*N*′-dimethylformamide (DMF) and couplings with 5:5:5:10 eqs of Fmoc-amino acid:activator:OxymaPure^®^:DIEA (activator was *O*-(benzotriazol-1-yl)*N*,*N*,*N*′,*N*′-tetramethyluronium hexafluorophosphate (HBTU for the first coupling) and *O*-(benzotriazol-1-yl)-*N*,*N*,*N*′,*N*′-tetramethyluronium tetrafluoroborate (TBTU) for the second coupling) in DMF. Peptides were cleaved with TFA:TIS:EDT:H_2_0 (92.5:2.5:2.5:2.5) (TIS is triisopropylsilane; EDT is 1,2-ethanedithiol).

Peptides were purified by RP-HPLC on a Water Corp XBridgeTM BEH C18 column (Water Corp, Milford, MA, USA) using 0%–70% ACN in water gradient at a flowrate of 1 mL/min for 30 min. 

The molecular masses of the synthetic peptides were corroborated by electrospray ionization mass spectrometry (ESI-MS) using Shimadzu LCMS-2020 equipment (Shimadzu Corp, Kyoto, Japan), in a 0%–100% ACN in water gradient for 20 min. Peptides were lyophilized and stored until use.

### 4.9. Secondary Structure Analysis

Circular dichroism (CD) spectroscopy was performed as described by Carvajal-Rondanelli et al. [43]. Briefly, CD spectra were carried out using a JASCO J-815 CD Spectrometer (JASCO, Corp., Tokyo, Japan) in the far ultraviolet (UV) range of 190–250 nm, using quartz cuvettes of 0.1 cm path length and 1 mm bandwidth. Each spectrum was the average of three scan repetitions in continuous scanning mode with 100 nm/min scanning speed with a response time of 2 s. Molar ellipticity was calculated with 1 mM solution of peptide, both in water and in 30% TFE. 

The CD spectra of each peptide was recorded at 37 °C, and the data were analyzed using the Spectra Manager software (version 2.0, JASCO Corp, Tokyo, Japan).

### 4.10. Peptide Modifications by Alanine Scan or Lysine Substitution

To identify the contribution of each amino acid residue to the peptide functionality, an alanine scan series of the 8 residues of the AQ-1766 peptide was synthesized and CD determined, as described previously; all the peptides were tested for antibacterial and cytotoxic activities. Identified functional amino acids by alanine scan were substituted by lysine residues, and these analog peptides were characterized and tested for antibacterial activity and cytotoxicity as well.

### 4.11. Cytotoxic Activity of Synthetic Peptides

Cytotoxic effect of both the original peptides and their analog sequences (alanine scan series and lysine modified sequences) on eukaryotic cells was evaluated by exposing human embryonic kidney (HEK293) cells (ATCC CLR-1573) to peptides resuspended in phosphate buffer saline (PBS) 1× at those concentrations tested for bactericidal activity, according to Sunarintyas et al [57]. Briefly, cell monolayers at 80% semi-confluence were washed with PBS and exposed for 1 h to peptides at concentrations ranging from 25 to 100 μM. Cytotoxicity was determined by MTS CellTiter One Solution cell proliferation (Promega, Madison, Wisconsin, USA) following the instructions of the manufacturer. 

Additionally, cytotoxic effect of the active peptides was analyzed by flow cytometry. For this, HEK 293 cells were exposed for 1 h to 25 µM of peptides and incubated for 1 min at room temperature with 200 µM of propidium iodide (PI) (Molecular Probes); then, cells were treated with trypsin, harvested by centrifugation, washed once with PBS 1×, and resuspended in the same buffer, and 10,000 cells were analyzed by flow cytometry CytoFLEX. All the experiments were performed two times in duplicate. Analyses were done using CytExpert 2.0 software (Beckman Coulter, Bea, CA, USA). 

Cells without peptides and incubated with the cytotoxic peptide aurein derived from the frog *Litoria aurea* [31] were used as negative and positive controls, respectively. All assays were carried out in triplicate.

### 4.12. Statistical Analysis

Graphics and statistical calculations were performed by the Graphpad Prism v6.1 for Windows, (GraphPad software, San Diego, CA, USA). Results were expressed as mean plus standard deviation and analyzed by one-way ANOVA. Those results showing significant differences were analyzed by Dunnett´s multiple comparison test. Significant differences at *p* < 0.05.

## 5. Conclusions

In this report, de novo sequences of three peptides with antibacterial activity from the microalga *T. suecica* were determined. One of these peptides, AQ-1766, which exhibited the higher activity, was subjected to an alanine scan and subsequently to lysine replacement of some residues. This procedure resulted in six peptides with improved antibacterial activity (AQ-3001, AQ-3002, AQ3369, AQ-3370, AQ-3371, and AQ-3372) with respect to the original peptide (AQ-1766).

Lysine replacements caused a modification of the secondary structure tendency of the peptides, with a change of beta turn tendency to helical trend, which is related to the electronic interaction of the cationic lysine residues with the aromatic residues of the peptide. 

To our knowledge, this is the first report showing the existence of antimicrobial peptides in microalgae without requiring enzymatic treatment, and the first report of antibacterial peptides in *T. suecica*.

## Figures and Tables

**Figure 1 marinedrugs-17-00453-f001:**
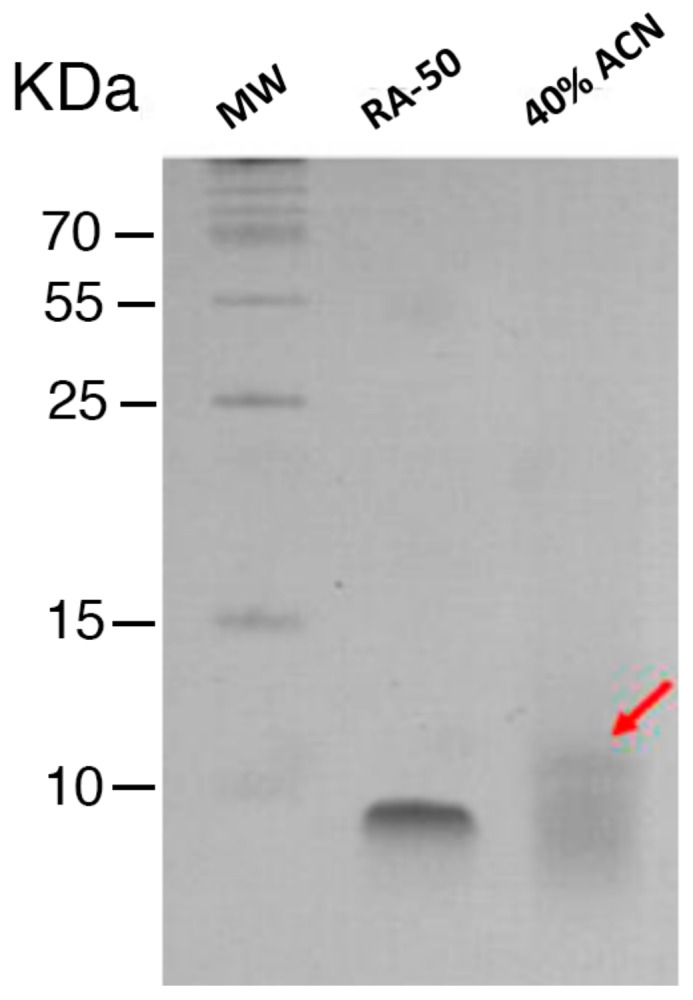
Electrophoretic protein profile of post Sep-Pak 40% Acetonitrile (ACN) eluted fraction from *Tetraselmis suecica*. Extract resolved through Tris-Tricine SDS-PAGE. Lane 1, molecular weight standards; lane 2, synthetic analog of trout cathelicidin RA-50 used as positive control [28]; lane 3, low molecular weight peptides (red arrow).

**Figure 2 marinedrugs-17-00453-f002:**
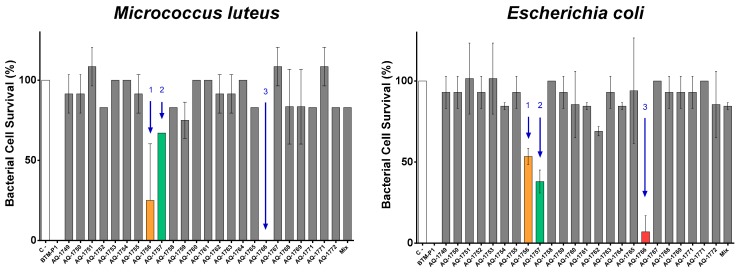
Antibacterial activity of chemically synthesized peptides identified in *T. suecica*. Bacterial survivals of *M. luteus* and *E. coli* after peptide exposition (50 µM) were analysed by MBC assay. C−: negative control bacteria without peptide; C+: positive control peptide BTM P1 from *B. thuringiensis*. Peptides with the highest antibacterial activity are indicated with arrows: AQ-1755 (1), AQ-1756 (2), and AQ-1766 (3).

**Figure 3 marinedrugs-17-00453-f003:**
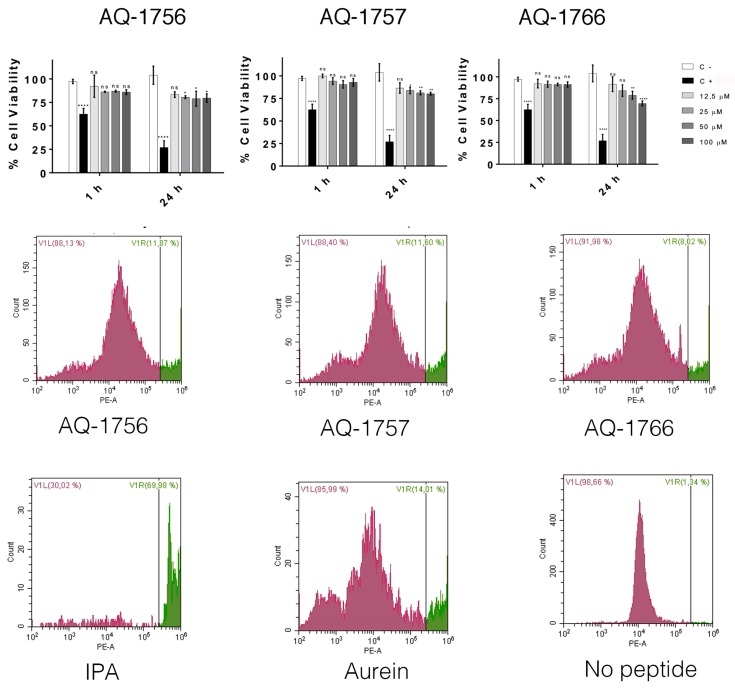
Cytotoxicity analysis of synthetic peptides. Viability of HEK293 cells after AQ-1756, AQ-1757, and AQ-1766 exposure was analyzed by MTS and flow cytometry (top and medium panels respectively) assays. In the MTS assay, each column represents a concentration; ns: no significant differences; significant differences: **** *p* < 0.0001, *** *p* < 0.001, ** *p* < 0.01, and * *p* < 0.05. In flow cytometry assays, viable and dead cells are presented in purple and green, respectively. In the lower panel, IPA (isopropanol) and Aurein (cytotoxic peptide from the frog *Litoria aurea*; [31]) used as positive controls (C+) and untreated cells (no peptide) as negative controls (C−) are shown.

**Figure 4 marinedrugs-17-00453-f004:**
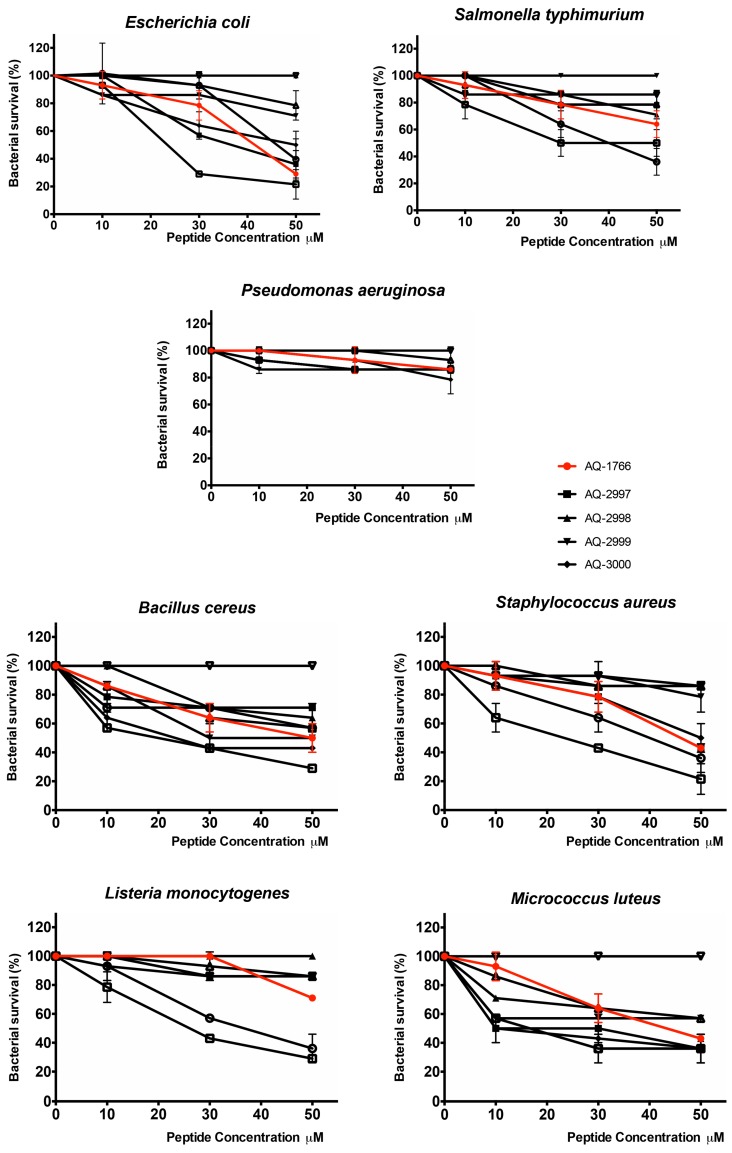
Antibacterial activity of alanine scan analogs of peptide AQ-1766. Bacterial survival after alanine scan series exposition at three peptide concentrations (10, 30, and 50 µM). Analysis was made by MBC assay. Percentage of survival was calculated with respect to the negative control (bacteria without peptide).

**Figure 5 marinedrugs-17-00453-f005:**
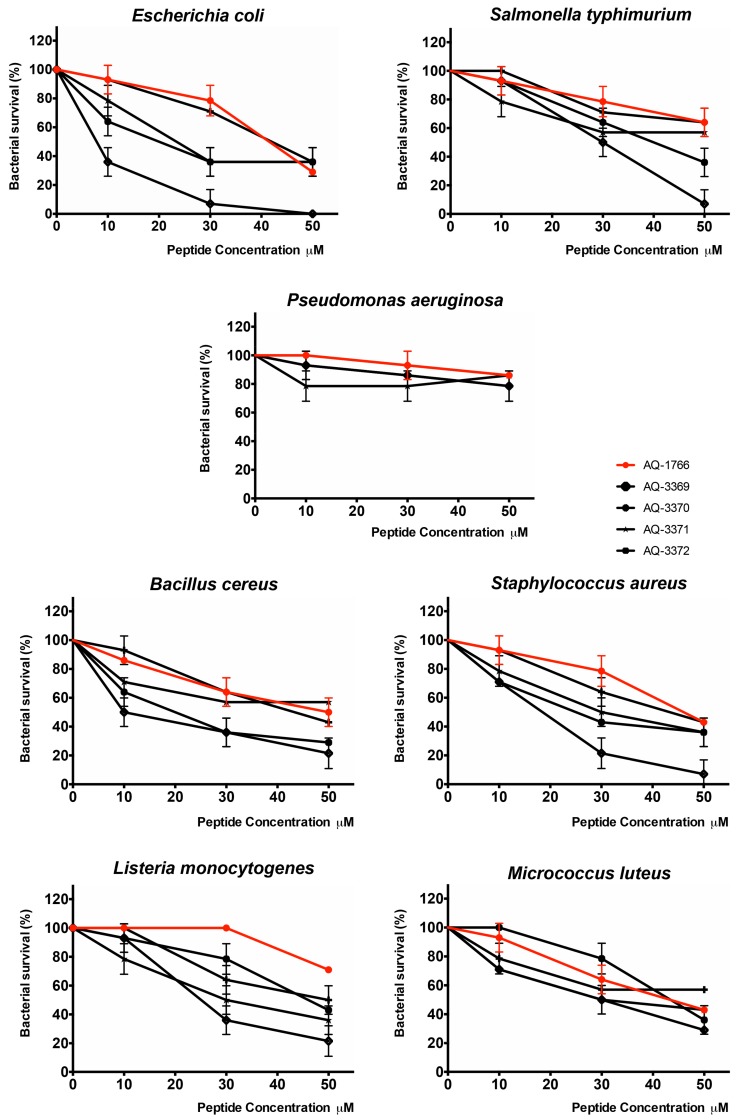
Antibacterial activity of AQ-1766 and its lysine analogs. Bacterial survival (%) after lysine scan series exposition at three peptide concentrations (10, 30, and 50 µM). Analysis was made by MBC assay. Percentage of survival was calculated with respect to the negative control bacteria without peptide.

**Figure 6 marinedrugs-17-00453-f006:**
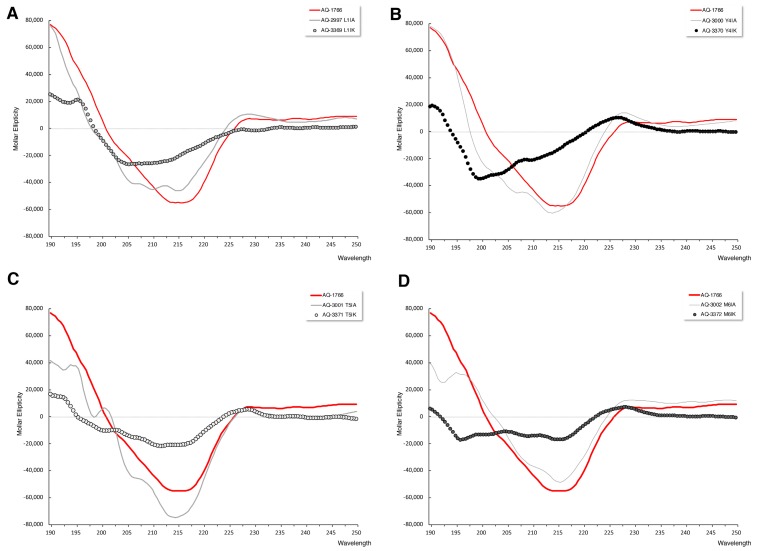
Circular dichroism of the peptide AQ-1766 and its alanine and lysine analogs. CD spectra of peptides were recorded at 1 mM concentration and 30% trifluoroethanol (TFE). Original peptide AQ-1766 spectrum is represented by red lines, alanine susbtituted peptides by grey lines, and lysine susbtituted peptides by dotted lines. (**A**) CD spectra of AQ-1766 and peptides with leucine residues replaced by lysine (AQ-3369) or alanine (AQ-2997) in the first position. (**B**) CD spectra of AQ-1766 and peptides with tyrosine residue exchange by lysine (AQ-3370) or alanine (AQ-3000) in the fourth position. (**C**) CD spectra of AQ-1766 and threonine residue exchange by lysine (AQ-3371) or alanine (AQ-3001) in the fifth position. (**D**) CD spectra of AQ-1766 and peptides with methionine residue exchange by lysine (AQ-3372) or alanine (AQ-3002) in the sixth position.

**Table 1 marinedrugs-17-00453-t001:** Antibacterial activity of the peptides present in the 40% ACN fraction derived from acid extracts of *Tetraselmis suecica* against Gram− and Gram+ bacteria.

Bacteria		Bacterial Cell Survival (%) ^a^			
		40% ACN Peptide Concentration			BTM P1 ^b^ (C+) 0.06 μg/μL
		0.1 μg/μL	0.3 μg/μL	0.5 μg/μL	
**Gram−**	*Escherichia coli* ML35	56	23	4	0
	*Salmonella typhimurium* ATCC 14028	93	82	78	7
	*Pseudomonas aeruginosa* ATCC 27853	100	93	93	11
**Gram+**	*Bacillus cereus* ISP B7/13	52	0	0	0
	*Staphylococcus aureus* ATCC 25933	89	48	19	4
	*Listeria monocytogenes* ATCC 19115	100	82	93	11
	*Micrococcus luteus* ATCC 9341	100	96	96	7

^a^ Bacterial survival of Gram– and Gram+ bacteria exposed to the 40% acetonitirle (ACN) eluted fraction was evaluated by the minimal bactericidal concentration (MBC) assay. Cultures containing 1 × 10^7^ CFU/mL were exposed for 1 h at 37 °C to different concentrations of extracts; then, bacteria were subjected to seven ten-fold serial dilutions and incubated 16 h at 37 °C. Surviving bacteria were quantified as CFU/mL and compared with bacterial growth without treatment. ^b^ Antimicrobial peptide BTM P1 derived from *Bacillus thuringiensis* was used as positive control. Highest activities (survival < 50%) are highlighted in bold characters.

**Table 2 marinedrugs-17-00453-t002:** Analog peptides of AQ-1766, with replacement of alanine or lysine as indicated in bold characters.

Alanine Analogs		Lysine Analogs	
Peptide	Sequence	Peptide	Sequence
AQ-1766	LWFYTMWH	AQ-1766	LWFYTMWH
AQ-2997	**A**WFYTMWH	AQ-3369	**K**WFYTMWH
AQ-2998	L**A**FYTMWH		
AQ-2999	LW**A**YTMWH		
AQ-3000	LWF**A**TMWH	AQ-3370	LWF**K**TMWH
AQ-3001	LWFY**A**MWH	AQ-3371	LWFY**K**MWH
AQ-3002	LWFYT**A**WH	AQ-3372	LWFYT**K**WH
AQ-3003	LWFYTM**A**H		
AQ-3004	LWFYTMW**A**		

**Table 3 marinedrugs-17-00453-t003:** MBC_50_ for alanine scan peptides of AQ-1766. Those results showing significant differences with respect to the original peptide, analyzed by Dunnett´s multiple comparison test, are indicated in bold as follows: **** *p* < 0.0001, *** *p* < 0.001, ** *p* < 0.01, and * *p* < 0.05.

	Peptide	AQ-1766	AQ-2997	AQ-2998	AQ-2999	AQ-3000	AQ-3001	AQ-3002	AQ-3003	AQ-3004
Bacteria										
Gram−	*E. coli ML35*	41	36	NA	>50	50	46	**23 *****	>50	NA
	*S. typhimurium ATCC 14028*	>50	>50	>50	NA	>50	40	30	>50	>50
	*P. aeruginosa ATCC 27853*	>50	>50	>50	>50	>50	>50	>50	>50	>50
Gram+	*B. cereus ISP B7/13*	50	>50	>50	**30 ****	**23 *****	>50	**20 ******	>50	NA
	*S. aureus ATCC 25933*	46	46	>50	>50	50	43	**23 *****	>50	>50
	*L. monocytogenes ATCC 19115*	>50	>50	>50	>50	>50	36	26	>50	>50
	*M. luteus ATCC 9341*	44	10	44	>50	**10 ***	**17 ***	**17 ***	>50	NA

NA = no activity detected.

**Table 4 marinedrugs-17-00453-t004:** MBC_50_ for lysine analog peptides of AQ-1766. Those results showing significant differences with respect to the original peptide, analyzed by Dunnett´s multiple comparison test, are indicated in bold as follows: **** *p* < 0.0001, *** *p* < 0.001, ** *p* < 0.01 and * *p* < 0.05.

	Peptide	AQ-1766	AQ-3369	AQ-3370	AQ-3371	AQ-3372
Bacteria						
Gram−	*E. coli ML35*	41	**8 ******	**20 *****	**23 *****	42
	*S. typhimurium ATCC 14028*	>50	30	40	>50	>50
	*P. aeruginosa ATCC 27853*	>50	>50	>50	>50	>50
Gram+	*B. cereus ISP B7/13*	50	**10 ******	**20 ******	>50	43
	*S. aureus ATCC 25933*	46	**19 *****	**25 ****	**30 ***	44
	*L. monocytogenes ATCC 19115*	>50	25	46	30	50
	*M. luteus ATCC 9341*	44	30	44	30	>50

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
