# Peer review of "Identification of Antimicrobial Peptides from the Microalgae Tetraselmis suecica (Kylin) Butcher and Bactericidal Activity Improvement"

_marinedrugs, 2019, doi:10.3390/md17080453_

Round 1

Reviewer 1 Report

Lines 124-126, the sentence "None of the three peptides showed a significant difference with respect to the untreated cells at the lowest tested concentration" needs to state expressly that no significant difference was detected after 1 hour of incubation (because at longer incubation peptides have shown some activity in this test).

On Fig. 3 or in its legend, please provide proper designations of the top panel bar diagrams: which colour means which substance and at what concentration. This applies mostly to white (probably untreated cells) and black (IPA? Aurein?) bars.

Lines 126-127 "AQ-1766 showed the lowest toxicity" - I don't see data that proves this statement, probably because Fig.3 lacks proper legends.

On Fig. 4, it would be better to add some statistical analysis showing the level of bacterial growth inhibition significance. Same applies to Fig. 5.

Line 161, "substitutions had no significative effect". What statistical analysis has been performed to show the lack of significance? And to what data analysis was applied: MBC50 or %? Figures should express this explicitly, which is more convenient for the readers.

It would probably be more informative to add 95% confidence intervals to MBC50 to illustrate the significance statement. OK, data on figures and tables are interconnected so the statistical analysis could be depicted either on the figures or in the tables (probably the second is better, but I'm not insisting).

Line 177, "substitution by alanine".

Author Response

Lines 124-126, the sentence "None of the three peptides showed a significant difference with respect to the untreated cells at the lowest tested concentration" needs to state expressly that no significant difference was detected after 1 hour of incubation (because at longer incubation peptides have shown some activity in this test).

Response

First we apologize to the reviewer, the confusion is caused by the lack of designations on Figure 3, an error that he points out in the next point. 

According to the results presented in Figure 3, now corrected, there are no differences for the three peptides at all the used concentrations within one hour, respect to the untreated cells, which is stated in line 126; and in addition at 24 h there are no significant difference with the lowest concentration used, 12,5 mM for the three peptides, which is the sentence mentioned by the reviewer which is in lines 128-130.

On Fig. 3 or in its legend, please provide proper designations of the top panel bar diagrams: which colour means which substance and at what concentration. This applies mostly to white (probably untreated cells) and black (IPA? Aurein?) bars.

Response

It was a mistake in the construction of the graph. Now it is included. 

Lines 126-127 "AQ-1766 showed the lowest toxicity" - I don't see data that proves this statement, probably because Fig.3 lacks proper legends.

Response

Figure 3 is now complete with the corresponding labels. In the MTS assay the cytotoxicity was similar for the three peptides, only slightly lower for the AQ-1766; however, in the flow cytometry assay the viability was 88.13%, 88.4% and 91.98% for the peptides AQ-1756, AQ-1757 and AQ-1766 respectively (lines 133-135 and figure 3 in the cell count included in the medium panel, upper left corner in purple letters).

On Fig. 4, it would be better to add some statistical analysis showing the level of bacterial growth inhibition significance. Same applies to Fig. 5. 

Line 161, "substitutions had no significative effect". What statistical analysis has been performed to show the lack of significance? And to what data analysis was applied: MBC50 or %? Figures should express this explicitly, which is more convenient for the readers.

It would probably be more informative to add 95% confidence intervals to MBC50 to illustrate the significance statement. OK, data on figures and tables are interconnected so the statistical analysis could be depicted either on the figures or in the tables (probably the second is better, but I'm not insisting).

Response

Thanks for the remark. The statistical analysis is now indicated in the tables, and was calculated from the raw data before obtaining the average. The significance is indicated in tables 3 and 4 at  P < 0.0001, ***P < 0.001, ** P < 0.01 and * P < 0.05. Some of the data could not be analyzed in this context, since it was not possible to determine the MBC50with the concentrations used in the assays. This is also explained in the text.

Lines  173-175, and 196-198 correspond to the captions of Tables 3 and 4: “Those results showing significant differences respect to the original peptide, analyzed by Dunnett´s multiple comparison test are indicated in bold as follows: **** P < 0.0001, ***P < 0.001, ** P < 0.01 and * P < 0.05”.

Lines 156-158: “In addition, AQ-1766 showed low activity against S. typhimuriumP. aeruginosaand L. monocytogenesand it was not possible to determine MBC50at the concentrations used nor to perform the statistical analysis (Figure 4 and Table 3)”.

Lines 184-188: “As in the alanine scan, it was not possible to determine the significance of the difference against S. typhimuriumand L. monocytogenesof the original versus lysine analog peptides, but a reduction in the MBC50was observed. In the case of P. aeruginosa, although it was not possible to determine MBC50 at the used concentrations, the survival curves showed a decrease in bacterial survival percentage”.

Line 177, "substitution by alanine".

Response

Done (line 184)

Reviewer 2 Report

Guzman et al report on the analysis of acid extracts from the microalga Tetraselmis suecica that showed antibacterial activity. The active compounds were found to be peptides and their sequence was determined using de novo sequences. Synthetic peptides and their alanine and lysine analogs allowed identifying key residues influence their antibacterial activity

Throughout the manuscript:

-bacterial strain names are not written in italic (except tables)!!

- space between SI untis such as °C and h

Introduction:

The introduction is written in a very confusing way and the sections appear in a non-logical order

Please revise introduction to improve the text flow and logic: e.g.

1.       Line 44-59: it appears to me that this section should come first within the introduction or after line 37

2.       Line 38-43: put sentences into logical order; sentence of line 42-43 need to follow first sentence of this section

Results:

The first section starts directly into the topic; no explanation given why the 40% ACN fraction was evaluated. Although most parts are written in the experimental section, it is essential to start this section with some more details why the authors have chosen this approach.

In section 2.2, please add the following informations:

-           how the purification was performed (shortly mention at least the gradient used)

-          Add a short comment how these peptides were synthesized

Table 1. shows very high activity of enriched fraction; which is several fold higher (0.5 ug/uL) compared to the isolated or modified peptides. It is likely that these high acitivties are due to additive or synergistic effects of the isolated peptides. I would strongly encourage to perform combination assays (combination of synthesized peptides tested against on representative test strain) to interlink the ecological relevant/observed activity and the AMP identification.

Please revised the text within the Discussion:

-          line 2019-221:  example of cyanobacteria is misleading; no connection to algae

-          include comments why the observed activity was much lower compared to the isolated peptides

Author Response

Throughout the manuscript:

-bacterial strain names are not written in italic (except tables)!!

Response

Now all bacterial names are in italics

- space between SI untis such as °C and h

Response

Done. 

Introduction:

The introduction is written in a very confusing way and the sections appear in a non-logical order

Please revise introduction to improve the text flow and logic: e.g.

1.       Line 44-59: it appears to me that this section should come first within the introduction or after line 37

2.       Line 38-43: put sentences into logical order; sentence of line 42-43 need to follow first sentence of this section

Response

Thanks to the reviewer for the observations. The introduction was reorganized and modified. Please see lines 38-67.

Results:

The first section starts directly into the topic; no explanation given why the 40% ACN fraction was evaluated. Although most parts are written in the experimental section, it is essential to start this section with some more details why the authors have chosen this approach

Response

As the reviewer points out, although most of the justification is found in the methods section (lines 322-324), we did not include previous work by our group, where this procedure has already been used with good results.  Now we have added a short explanation in section 2.1 

Lines 70-74: Microalga was submitted to acid extract, and purified by C-18 reverse-phase chromatography to obtain fractions using an elution gradient of 5, 40, 80 and 100% acetonitrile (ACN) in 0.01% trifluoroacetic acid (TFA) in water. This procedure modified from Mitta [25], has been used by our group in the search of peptides from other natural products, with good results [26,27]. 

In section 2.2, please add the following informations:

-           how the purification was performed (shortly mention at least the gradient used)

Response

Lines 99-103: The 40% ACN eluted fraction was further purified until homogeneity by reverse phase high performance liquid chromatography (RP-HPLC) in a continuous 0-80% ACN gradient in 0.01% TFA in water for 20 min at 1 mL/min. HPLC fractions were collected, lyophilized and subsequently characterized by MALDI-TOF mass spectrometry and by MS/MS mass spectrometry to determine de novo sequences of the peptides.

-          Add a short comment how these peptides were synthesized

Response

Lines 109-110: The identified peptides were then chemically synthesized by Fmoc solid-phase strategy with amidated C-terminal and their antimicrobial properties were evaluated. 

Table 1. shows very high activity of enriched fraction; which is several fold higher (0.5 ug/uL) compared to the isolated or modified peptides. It is likely that these high acitivties are due to additive or synergistic effects of the isolated peptides. I would strongly encourage to perform combination assays (combination of synthesized peptides tested against on representative test strain) to interlink the ecological relevant/observed activity and the AMP identification.

Response

The results shown in Table 1 are expressed as µg/µL, and the results of the peptides are expressed as µM concentration; when transforming the concentrations of the peptides used, to express them in the same units, concentrations obtained are between 0.01 µg/µL (10 µM) and 0.06 µg/µL (50 µM), which are considerably lower than the concentration in the extracts. It is possible that with higher peptide concentrations similar results to the extract will be achieved. However, with peptides it is better to look for the lowest active concentrations to avoid toxicity, and also to have a high cost-benefit ratio. 

We agree with the reviewer in the fact that the activity of the 40% ACN extract could be due to a synergistic effect or also to the presence of another compounds, such as secondary metabolites. The combined peptide assays recommended by the reviewer are considered within the projections of the group`s work, as an alternative to cover a broader spectrum of bacteria.

Please revised the text within the Discussion:

-          line 2019-221:  example of cyanobacteria is misleading; no connection to algae

Response

Sorry for this misleading. Cyanobacteria were formerly classified as microalgae, as in the case of the cited reference. Now this has been amended as indicated below.

Lines 231-233: “The antibacterial properties of protein hydrolysates were described for the microalgae Dunaliella salina[35]and Chlorella sorokiniana[36]but the specific peptides responsible for this activity were not determined.”

-          include comments why the observed activity was much lower compared to the isolated peptides

Response

With respect to the phrase "the activity observed was much smaller" we do not understand if the reviewer is referring to the synthesized peptides activity. In the work we did not measure the activity of the isolated peptides, we simply determined the activity of the extract, and from this, the de novosequence of the peptides present. These peptides were synthesized to determine those with higher activity.

We included a paragraph regarding this issue in the discussion section commenting about the difference of the activity of the extract with respect to the synthesized peptides.

Lies 285-289: The activity of the 40% ACN extract seems to be higher than that of the synthesized peptides (Table 1); however, the concentration of the peptides used ranged from 0.01 µg/µL (10 µM) to 0.06 µg/µL (50 µM), for the different peptides, which is at least one order of magnitude lower, than in the extract. In addition to this having a pure peptide whose structure is known, allows the assays to be easily reproducible and scalable, which does not occur with natural extracts.